# Nitrogen Preference of Dominant Species during Hailuogou Glacier Retreat Succession on the Eastern Tibetan Plateau

**DOI:** 10.3390/plants12040838

**Published:** 2023-02-13

**Authors:** Yulin Huang, Liushan Du, Yanbao Lei, Jiye Liang

**Affiliations:** 1China-Croatia “Belt and Road” Joint Laboratory on Biodiversity and Ecosystem Services, CAS Key Laboratory of Mountain Ecological Restoration and Bioresource Utilization & Ecological Restoration and Biodiversity Conservation Key Laboratory of Sichuan Province, Chengdu Institute of Biology, Chinese Academy of Sciences, Chengdu 610041, China; 2University of Chinese Academy of Sciences, Beijing 100049, China; 3School of Pharmacy, Youjiang Medical University for Nationalities, Baise 533000, China

**Keywords:** nitrogen uptake preference, primary succession, glacier retreat area, nitrogen isotopes, ammonium, nitrate, soluble organic N, dominant plants

## Abstract

Plant nitrogen (N) uptake preference is a key factor affecting plant nutrient acquisition, vegetation composition and ecosystem function. However, few studies have investigated the contribution of different N sources to plant N strategies, especially during the process of primary succession of a glacial retreat area. By measuring the natural abundance of N isotopes (δ^15^N) of dominant plants and soil, we estimated the relative contribution of different N forms (ammonium-NH_4_^+^, nitrate-NO_3_^−^ and soluble organic N-DON) and absorption preferences of nine dominant plants of three stages (12, 40 and 120 years old) of the Hailuogou glacier retreat area. Along with the chronosequence of primary succession, dominant plants preferred to absorb NO_3_^−^ in the early (73.5%) and middle (46.5%) stages. At the late stage, soil NH_4_^+^ contributed more than 60.0%, In addition, the contribution of DON to the total N uptake of plants was nearly 19.4%. Thus, the dominant plants’ preference for NO_3_^−^ in the first two stages changes to NH_4_^+^ in the late stages during primary succession. The contribution of DON to the N source of dominant plants should not be ignored. It suggests that the shift of N uptake preference of dominant plants may reflect the adjustment of their N acquisition strategy, in response to the changes in their physiological traits and soil nutrient conditions. Better knowledge of plant preferences for different N forms could significantly improve our understanding on the potential feedbacks of plant N acquisition strategies to environmental changes, and provide valuable suggestions for the sustainable management of plantations during different successional stages.

## 1. Introduction

Nitrogen (N) is a requisite mineral element for plants and is generally considered to be the main factor limiting plant growth, reproduction and development in terrestrial ecosystems [1,2]. Ammonium (NH_4_^+^) and nitrate (NO_3_^−^) in soils are two important inorganic sources used by plants [3]. In addition, plants also absorb small amounts of dissolved organic nitrogen (DON) such as amino acids [4]. The N availability and the selectivity of uptake preference are shown to be one of the important factors in determining ecosystem productivity, structure and function, and ecological succession [5].

Some studies have confirmed three categories in preference of different N forms. (1) Preference for NO_3_^−^. For example, plants in temperate semi-arid steppe prefer nitrate over ammonium and glycine [6]. (2) Preference for NH_4_^+^. *Cunninghamia lanceolata* of different ages showed a preference for NH_4_^+^, and little uptake of NO_3_^−^ and glycine [7]. (3) Preference for DON. In N-restricted ecosystems such as Arctic tundra, boreal forest, alpine meadow, temperate forest and temperate grassland, plants have a preference for soil amino acids even higher than inorganic N [7,8,9]. In addition, there are seasonal differences in N preferences among the same species. *Leymus chinensis*, the dominant plant in the Inner Mongolia grassland, prefers NO_3_^−^ in January and NH_4_^+^ in August [10]. The main reason for the difference lies in the complexity of biotic and abiotic factors affecting different N forms in soil, including plant functional characteristics, mycorrhizal fungi and soil nutrient status, and other environmental factors [11,12]. Generally, soil nutrient status is an important factor in plant N uptake and utilization strategies, as plants usually take advantage of the dominant N resources in soil [13,14]. N uptake by plants is also affected by climate conditions. When temperature decreases, N acquired by plants can shift from an inorganic to an organic form [15]; meanwhile, an increase in the soil moisture will render plants to switch their uptake from NO_3_^−^ to NH_4_^+^ [16,17]. When plants have symbiotic mycorrhizal fungi, they can also directly absorb organic N from soil [18]. In addition, soil pH also affects N preference, i.e., NH_4_^+^ is more likely to be absorbed by tree species growing in acidic soil, compared with NO_3_^−^ in medium or alkaline soil [19]. These results evidently demonstrate the plasticity of plant N use strategies, which means that when plants are confronted with the interaction of multiple factors, they often choose the most appropriate strategies to meet their own growth and development needs.

Previous studies on plant N preference were mainly collected from greenhouse experiments or field isotope labeling [15,20]. The natural abundance of ^15^N impacted by the exchange and circulation of N in the soil may be more straightforward [21]. A growing number of studies have used the natural abundance of ^15^N in soils and plants (δ^15^N) to reflect and predict plant N utilization characteristics and preferences [22,23]. Foliar δ^15^N values have been demonstrated as being close to the δ^15^N of their N sources, and can be used to integrate N availability in forest ecosystems [22,24]. By analyzing N contents and natural abundance values of ^15^N in different forms, N uptake preferences and their relative contributions can be assessed based on the isotope mixing model. This method has no artificial N-addition and does not change size of the soil N pool to affect plant N uptake [25,26]. For example, Takebayashi et al. [24] studied the utilization of NH_4_^+^ and NO_3_^−^ by plants at different N availability gradients by measuring the natural abundance of ^15^N in leaves and soil. Their results showed that Japanese cypress (*Chamaecyparis obtusa*) mainly used NH_4_^+^ in N-restricted forests, while plants in N-rich areas were more dependent on NO_3_^−^.

The acceleration of glacier melting caused by global warming in recent years has created bare land in different periods which can be accurately measured, providing ideal places to study the primary succession of vegetation [27,28,29]. Most studies on vegetation succession in glacial retreat areas focus on the vegetation composition and structure, soil nutrient status and the coupling relationship of plant-soil-microbial interactions [30,31,32,33]. However, the N utilization strategies of dominant plants at different stages of primary succession are poorly understood, and therefore, the patterns and mechanisms for plants adjusting their N preference to adapt to environmental changes remain to be further studied.

The Hailuogou Glacier Chronosequence, located on the eastern fringe of the Tibetan Plateau, is an ideal place for exploring the correlations between N use strategies and N status across the primary succession, as the mild and humid conditions promote the rapid accumulation of organic matter, accelerated soil development and strong plant-soil feedbacks [34]. Along the approximately 2 km belt, sequential sites encompassing three stages of forest succession can be easily identified, extending from early pioneer N_2_-fixing grasses and shrubs, a diverse deciduous broad-leaved forest at the middle stage, to a late climax evergreen community stage [34,35]. The dominant N forms in the soil change from NO_3_^−^ to NH_4_^+^ due to the strong mobility of NO_3_^−^; meanwhile, the organic N increases along the chronosequence [34,35]. In this study we measured the natural abundance of ^15^N in the soil and foliage of dominant plants. By using a mass balance equation to quantitatively estimate the source proportion of foliar N, we studied the contribution of N sources of NH_4_^+^, NO_3_^−^ and DON in the soil at different stages of the complete forest primary succession. We hypothesized that (1) the preference of dominant plants for NO_3_^−^ gradually changed to NH_4_^+^ with the primary succession, and (2) the changing preference was affected by the interaction of many environmental and physiological factors. The study of plant N preferences and utilization strategies can deepen our understanding of plant N acquisition and provide some theoretical support for clarifying the mechanism of plant coexistence, competition and community assembly.

## 2. Results

### 2.1. Soil Physicochemical Properties and N Contents of Different Forms at the Primary Succession

Along the primary succession, there was no significant difference in soil pH between the early and middle stages (*p* > 0.05), but the soil pH decreased significantly at the late stage (*p* < 0.05; Table 1). Soil organic matter (SOM) content increased significantly and the average SOM contents in the early, middle, and late stages were 1.50, 2.30 and 3.50 g·kg^−^^1^, respectively (Table 1). The soil dissolved inorganic nitrogen (DIN) content in the early and middle stages were significantly lower than that of the late stage (Table 1). Specifically, the NH_4_^+^ and NO_3_*^−^* contents gradually increased, with the highest value being recorded at the late stage (51.46 mg·kg^−^^1^, 19.41 mg·kg^−^^1^) (Table 1). The contents of total dissolved N (TDN) and DON also increased significantly at the late stage of the chronosequence (Table 1).

### 2.2. The Soil and Foliar δ^15^N Changes at the Primary Succession

Different changing patterns in the δ^15^N of the three N forms with the successional stages were observed (Figure 1). The δ^15^N-NH_4_^+^ value of the soil in the early stage was significantly higher than that in the middle and late stages (p < 0.05; Figure 1a). No significant difference was observed in the δ^15^N-NO_3_^−^ value between different stages (*p* > 0.05; Figure 1a). Moreover, there was no significant difference in the δ^15^N-TDN value of the soil at different stages, while a significant decrease in the δ^15^N-DON value (Figure 1b). Considering the internal isotope fractionation effect, the modified foliar δ^15^N values were close to the N source of the dominant plant. Foliar δ^15^N values decreased from the early stage to the late stage, ranging from 0.36‰ to −2.95‰ (p < 0.05; Figure 2). The difference of foliar δ^15^N was 6‰ with NO_3_^−^ δ^15^N, 12‰ with NH_4_^+^ δ^15^N and 37‰ with DON δ^15^N in the middle period. The foliar δ^15^N value was within 7‰ with NO_3_^−^ δ^15^N, while it differed from 11‰ of NH_4_^+^ δ^15^N and 12‰ with DON δ^15^N in the later stage (Figure 2).

### 2.3. The Contribution of Different N Sources at the Primary Succession

If NH_4_^+^ and NO_3_^−^ in soil were the two N sources, there were significant differences in the contributions of NH_4_^+^ and NO_3_^−^ at each stage (*p* < 0.05; Figure 3a). In the first two stages, the dominant plants mainly assimilated NO_3_^−^ as the main N source, and the mean values of f_NO3_^−^ were 83.50% and 64.33%, respectively (Figure 3a). Instead, the contribution of NH_4_^+^ was the greatest in the later stage, with the mean f_NH4_^+^ (87.05%) significantly higher than that of f_NO3_^−^ (12.95%) (Figure 3a). Thus, significant increases in NH_4_^+^ contribution, and remarkable reductions in NO_3_^−^ were observed across the primary succession (Figure 3a). In addition, NH_4_^+^ content in soil was positively correlated with the contribution of NH_4_^+^ (R^2^ = 0.85, *p* < 0.01; Figure 4), indicating that the soil N content was an important factor affecting the contribution of plant N sources during primary succession.

When DON was included in the calculation, NO_3_^−^ still had the highest contribution in the early and middle stages, and the mean f_NO3_^−^ (73.50% and 46.50%) was significantly higher than f_NH4_^+^ (22.00% and 42.00%) and f_DON_ (4.50% and 11.50%), respectively (Figure 3b). However, no significant difference was observed between f_NH4_^+^ and f_NO3_^−^ in the middle stage (Figure 3b). Similarly, f_NH4_^+^ significantly increased to the highest (60.00%) at the later stage, and there was no significant difference in the contribution of NO_3_^−^ and DON (20.67% vs. 19.33%) (Figure 3b). During the succession stage, the acquisition of main N sources by dominant plants changed from NO_3_^−^ in the early stage to NH_4_^+^ in the late stage, and the contribution of DON gradually increased (from 4.50% to 19.44%) (Figure 3b). For dominant species, NO_3_^−^ made the largest contribution to the N uptake of Astragalus membranaceus and Hippophae rhamnoides in the early stage, while NH_4_^+^ had a greater contribution to the N absorption of Abies fabri, Picea brachytyla and Rhododendron simsii in the climax coniferous forests of the later stage (Figure 3).

### 2.4. Plants’ Preference for Different Soil N Forms at the Primary Succession

When only DIN was considered (the two-source mixing model), the results showed negative values of *β*_NH4_^+^ (−0.43 and −0.38) in the first two stages, but the value of *β*_NH4_^+^ was positive (0.27) in the late stage of succession (Figure 5a). This finding suggested that the N uptake preference of dominant plants for NO_3_*^−^* in the early and middle stages changes to NH_4_^+^ in the late stages. If DON was used as the third N source (the three-source mixing model), dominant plants showed similar N preferences. The *β*_NH4_^+^ values of the dominant plants during three stages were −0.20, −0.19 and 0.14, respectively, yet the *β*_DON_ values were all negative (−0.24, −0.05 and −0.18) (Figure 5b)

## 3. Discussion

### 3.1. Change of δ^15^N in Soil and Plants in Different Stages

The natural abundance of ^15^N (δ^15^N) can be used to track the uptake of different N forms in soil by plants. In this study, the δ^15^N value of soil in the retreat area of the Hailuogou Glacier showed a decreasing trend with the primary succession (Figure 1). N_2_-fixing plants such as *A. adsurgens* and *H. rhamnoides* dominated the early stage, and the N isotope fraction dominated by the N_2_-fixing effect was small; therefore, the soil δ^15^N value was relatively higher. However, Hobbie et al. [36] found that soil δ^15^N values increased with the distance of the primary succession of Lyman glacier in Cascade Mountain, Washington, USA. The main reason was that the primary succession was severely limited by N concentrations and lacked N_2_-fixing plants during pedogenesis, which is quite different from Hailuogou Glacier.

Some studies found that the foliar δ^15^N value was inconsistent with the variation range of soil δ^15^N-DIN [37,38], which suggested the importance of fractionation during plant N uptake. We observed that the decline in the trend of the foliar δ^15^N value was smaller with the direction of primary succession (Figure 1 and Figure 2). The magnitude of fractionation during plant N uptake was related to the soil DIN content, mycorrhizal status, mycorrhizal types and other factors [36]. The decrease in foliar δ^15^N may reflect the greater dependence of dominant plants on mycorrhizal fungal transfer N [39]. In the early stage of succession, *A. adsurgens* and *H. rhamnoides* depended on the symbiotic rhizobia in the roots to convert N_2_ from the atmosphere into NH_3_, which was then absorbed by plants [40]. Because of the small amount of N isotopic fractionation during biological N fixation, the early foliar δ^15^N value was almost close to those of atmospheric N isotopes (0‰) [41]. In general, plants associated with ectomycorrhiza (ECM), and ericoid mycorrhiza (ErM) usually have lower foliar δ^15^N values than those with or without arbuscular mycorrhizal fungus (AMF) [36,39]. *A. fabri* and *P. brachytula* with ECM were dominant at the later stage, resulting in decreasing foliar δ^15^N values. In addition, plant species also influenced leaf δ^15^N, and leaves of broadleaved trees generally contained more δ^15^N than conifers [42,43], which was also verified in our study (Figure 2).

The δ^15^N entering plant leaves was close to the main N source of plants [17,23,44]. In this study, with the progress of succession, the δ^15^N of dominant plants ranged from −2.95‰ to 0.36‰, which was closer to the δ^15^N of soil NO_3_^−^ (−4.44‰ to −7.05‰), but lower than that of NH_4_^+^ (7.73‰ to 37.49‰). The result indicated that dominant plants were more likely to absorb NO_3_^−^ during this period, or that the isotope fractionation resulted from the increased dependence of plants on soil NO_3_^−^ absorption.

### 3.2. Nitrogen Preference of Dominant Species in Different Stages

NH_4_^+^ and NO_3_^−^ are two inorganic N forms; therefore, plants using different forms of organic and inorganic N may develop diversified N acquisition and utilization strategies to meet their N requirements [19]. In some cases, plants would show a preference for one form, which may be an important factor in determining and predicting plant distribution and interaction with other species [19]. In this study, we compared the results of the two-source mixing model with NH_4_^+^and NO_3_^−^, and the three-source mixing model which involved the inclusion of DON as a potential N source for plants. This indicated that the dominant plants could directly utilize soil DON, but the main N source was still DIN (NH_4_^+^ and NO_3_^−^) (Figure 4 and Figure 5). The uptake of DIN for plants was significantly higher than that of amino acids. Furthermore, the main N source and preference changed dramatically in different stages, from NO_3_^−^ in the early stage to NH_4_^+^ in the later stage (Figure 4 and Figure 5) and the contribution of DON gradually increased. In addition, the difference between δ^15^N in plant leaves and soil δ^15^N-DON was not consistent with the difference between soil NH_4_^+^ or NO_3_^−^ at almost all stages, which also indicated that these plants had little or no preference for DON. Similar results have also been found in alpine forests [22], where N source preferences of spruce switched from NO_3_^−^ (20 and 30 y) to NH_4_^+^ (more than 40 y), and DON contributed from 23% to 44% of N sources. Compared with the three end-member model, the contribution of DIN to the dominant plant N source was undoubtedly overestimated when only two DIN types were considered (Figure 4 and Figure 5). However, many studies have proved that plants growing at high latitudes, high altitudes and cold regions also have the ability to absorb organic nitrogen [7,8,9], because the movement of microbes were usually limited by low temperature in these areas, which was not conducive to the turnover of soil organic nitrogen. As the available inorganic nitrogen was in short supply, the contribution of DON (especially amino acids) to plant N sources was particularly important. This difference probably reflected that the relatively mild and humid conditions which promoted the rapid colonization of vegetation and soil development also led to a greater microbial turnover and exoenzyme activities, and thus, higher mineralization rate and accumulation of various N components, [31,34,35,45], which made it more favorable for plants to use DIN instead of DON. 

Although the reasons affecting plant N source preference are not entirely understood, we could explain the changes in N preference according to the plant species, growth status and soil N pool supply during the primary succession of the Hailuogou glacier retreat area. A growing number of studies have shown that plant uptake preferences for different N sources were related to the relative abundance of different N forms in the soil [6,11,18]. In the chronosequence stages of the Hailuogou glacier area, the relatively large pool of NH_4_^+^ than NO_3_^−^ (Table 1) in soil may be the most abundant N source to promote the utilization of dominant plants. However, dominant plants showed a preference for NO_3_^−^ in the early and middle stages of succession (Figure 5), indicating that NH_4_^+^ was not the optimal choice for their absorption at this time. Soil NH_4_^+^ content was dominant in the late stages, and the dominant plants preferred to absorb NH_4_^+^; moreover, soil NH_4_^+^ content and the contribution of soil NH_4_^+^ to the plant N source were significantly and positively correlated (Figure 4). Kielland et al. [46] used the ^15^N labeling method to find that DON uptake in the late succession (coniferous forest) was higher than that in the early succession (deciduous forest), possibly due to the change in the soil DON pool. Our study found that the absorption of soil DON by dominant plants in coniferous forests in the late period may also be closely related to the relatively large soil DON reservoir. With the change in the availability of different N forms, the preference of the dominant plants for N sources changed accordingly.

Mycorrhization status and mycorrhizal taxa are also reasons that impact the difference in preference uptake of DIN and DON in plants [47]. For amino acid absorption, host plants can rely on roots with amino acid transporters, or transfer by mycorrhizal fungi, due to their inability to freely penetrate the cell membrane [48,49]. Mycorrhizal absorption represented the main route of N acquisition, accounting for 66% of N uptake by plants [50]. Many studies have shown that AFM can help plants to enhance their access to DIN and DON [51,52,53], and ECM and ErM were able to decompose and absorb complex DON (such as amino acids) [54]. In this study, the dominant plants in the middle stage, such as *Populus purdomii* and *Salix rehderiana*, were AFM which preferred to absorb NO_3_^−^ in the soil. In the middle and late stages, the contribution of DON to the N source of the dominant plants increased (Figure 5b). Therefore, we speculated that this might be related to special coniferous species with mycorrhizal symbiosis; particularly, ECM *A. fabri*, *P. brachytyla* and ErM *R. simsii* were beneficial to increase uptake of DON.

Soil pH may also cause changes in plant preferences for N availability, uptake and assimilation [19,55]. The soil pH decreased significantly in the late succession period (Table 1), which may also be the reason why the contribution of soil NH_4_^+^ to the plant N source gradually increased to the dominant level. It is worth noting that N preference by dominant plants changed from NO_3_^−^ to NH_4_^+^ during the development of primary succession, which may also be related to the decreased toxicity sensitivity of dominant plants to NH_4_^+^ [56]. NH_4_^+^ can enter plant roots quickly, resulting in an obvious accumulation effect [57], which can cause toxic effects and inhibit the absorption of important cationic nutrients such as potassium [58,59], which will be harmful to the growth of plants, especially young trees. Thus, the dominant plants at the early and middle stages in the study region (which are usually young seedlings) preferred NO_3_^−^ to avoid possible NH_4_^+^ toxicity. Although the uptake and assimilation of NH_4_^+^ are more efficient than that of NO_3_^−^ [58], the toxicity of NH_4_^+^ accumulation in plant tissues has balanced the benefits of its uptake in some studies [60]. *Pseudotsuga menziesii* in the early succession had a higher sensitivity to NH_4_^+^ toxicity than *Picea glauca* in the late succession [61]. Collectively, in the retreat area of the Hailuogou glacier, plant N absorption preference and utilization strategies are combined by many factors to produce a comprehensive effect, which leads to the fact that plants eventually have one method observed for N acquisition.

## 4. Materials and Methods

### 4.1. Study Sites

The study site is located in the retreated area of the Hailuogou glacier on Gongga Mountain on the eastern fringe of the Tibetan Plateau, China (29°30′ to 30°20′ N, 101°30′ to 102°15′ E, 7556 m a.s.l). In this area, the maximum and minimum monthly mean temperature in July is 11.9 °C and in January is −4.4 °C, respectively, with a mean annual temperature of 3.8 °C [62]. The mean annual precipitation is about 2000 mm and most rainfall occurrs between June and October [34]. Hailuogou glacier has been developing a special vegetation successional sequence after 120 years of primary succession, with a horizontal length of about 2 km and a vertical drop of only 127 m (from 2982 m to 2855 m). Based on the details of our previous surveys in the Hailuogou Glacier chronosequence [34,35,45,63], we selected three different successional stages representing the early, middle and late stages of the primary succession. Specific information on nine common dominant species of three stages is shown in Table 2. In the early stage (~12 years), the dominant plants comprise N_2_-fixing pioneer plants, such as *A. membranaceus* and *H. rhamnoides*. These pioneer species are quickly followed by the dominant species of *P. purdomii*, *S. rehderiana*, *Betula utilis* and *Salix magnifica* in a broad-leaved forest in the middle period (~40 years). At the later stage (~120 years), the dominant species of *A. fabri*, *P. brachytyla* and *R. simsii* were involved in the formation of the climax coniferous forest which dominates later succession (Table 2).

### 4.2. Sampling Collection

We set three replicate 5 m × 5 m plots > 15 m apart from one another at each chronosequence stage (except for the early stages, with 2 × 2 m plots and a 5 m distance between plots considering the small transect space). For plant samples, we randomly selected 5 representative trees during the growing season from each plot, and three leaves were selected from each mature tree, which should be annual plants and develop in full sunlight. The soil sample (0–15 cm) was collected from each plot by using a 5 cm diameter soil drill before mixing four points and the center of each plot completely to acquire one composite soil sample. The composite samples were passed through a 2 mm sieve after removing large stones, plant roots, and litter, which were put into sterile sealed bags with labels, stored in freezer and transported back to the laboratory.

### 4.3. Experimental Analysis

Soil pH and was measured using a pH meter (pH-Conductivity Meter, Leici Ltd., Shanghai, China) from the soil suspensions using a soil/water mass ratio of 1:10. Soil bulk density (BD) was determined by the weight of the air-dried soil samples and the volume of the cutting ring (200 cm^3^). SOM and soil TN were measured by a Vario MAX CN Element Analyzer (Elementar Analysensysteme GmbH, Hanau, Germany). The contents of NH_4_^+^ and NO_3_^−^ in soil were extracted by 2 M KCl. In brief, 6 g fresh soil samples were taken, 30mL 2 M KCl solution was added and shaken for 1.5 h. We obtained the soil extract by qualitative filter paper. A portion from the samples was immediately placed in a freezer tube and stored at −20 °C to analyze the δ^15^N value of different forms in soil. DIN (DIN = NH_4_^+^-N + NO_3_^−^-N) and TDN contents were measured by continuous flow analyzer (SEAL Analytical, Germany) from the rest. TDN concentrations were determined using the alkaline persulfate digestion method after measuring NO_3_^−^. The contents of DON depended on the difference between TDN and DIN (DON = TDN-DIN).

After washing the leaves with deionized water and drying them in an oven at 50 °C, we weighed all foliar and soil milled samples (0.3–0.35 mg) through a 100-mesh screen and then packed every sample in a 6 × 4 tin cup. The N isotope measurement was analyzed by the Institute of Applied and Ecological Sciences, Shenyang, Chinese Academy of Sciences. The total δ^15^N values in these milled samples were determined using a vario MICRO cube elemental analyzer (Elementar Ltd., Hanau, Germany) coupled to a stable isotope ratio mass spectrometer (IsoPrime100, IsoPrime Ltd., Manchester, UK). The carrier gas He flow rate was set to 200 mL/min with a reaction tube temperature of 950 °C and a reduction tube temperature of 600 °C. The δ^15^N calculation equation was as follows:δ^15^N (‰) = [R_Sample_/R_Standard_ − 1 × 1000](1)
where R_sample_ and R_standard_ represent the ^15^N/^14^N values of the sample and standard and the accuracy of δ^15^N value is ±0.25‰.

The values of δ^15^N-NH_4_^+^ and δ^15^N-NO_3_^−^ in soil extracts were determined based on the modified methods of isotopic analysis of nitrous oxide (N_2_O), which has been described in detail in Liu et al. [26] and Tu et al. [64]. To be specific, NH_4_^+^ is oxidized to NO_2_^−^ by alkaline hypobromite (BrO^−^), which is then quantitatively converted to N_2_O by hydroxylamine (NH_2_OH) in a strong acid condition. NO_3_^−^ is reduced to NO_2_^−^ by Cd powder and then converted to N_2_O by sodium azide (NaN_3_) in acetic acid buffer. The resulting instrument is analyzed in an N_2_O cryogenic system and isotope ratio mass spectrometer (IsoPrime100, IsoPrime Ltd., Manchester, UK).

### 4.4. Calculations

#### 4.4.1. Contributions of Different N Forms to Plant N Sources

We assumed that N loss between plants roots (underground) and leaves (aboveground) was equal, and that the δ^15^N value of plant leaves should be only half of the fractionation effect of plant isotopes [17]. Therefore, the δ^15^N values of dominant plant symbiosis with ectomycorrhizal fungus (EMF) and arbuscular mycorrhizal fungus (AMF) were corrected according to the methods of Zhang et al. [22] and Hu et al. [65].

We calculated two sets of the N isotopic mass balance equation respectively and quantified the relative contribution of each N form to the N source of dominant plants by determining the proportion of foliar N (*f*), composed of different N sources. NH_4_^+^ and NO_3_^−^ were assumed to be two main N source of dominant plants, which was calculated according to the method proposed by Liu et al. [11]:* δ^15^N_foliar_ = (δ^15^N-NH_4_^+^ × *f*_NH4_^+^ + δ^15^N-NO_3_^−^ × *f*_NO3_^−^)(2)
1 = *f*_NH4_^+^ + *f*_NO3_^−^
(3)
where *f*_NH4_^+^ and *f*_NO3_^−^ respectively represented the relative contribution of NH_4_^+^ and NO_3_^−^ to the N source of dominant plants, *δ^15^N_foliar_ was the corrected value of δ^15^N of dominant plants and δ^15^N-NH_4_^+^ and δ^15^N-NO_3_^−^ were the measured values in soil.

Except for NH_4_^+^ and NO_3_^−^, we assumed DON as an additional N source absorbed by dominant plants, and it was calculated referring to the method of Zhang et al. [22].
* δ^15^N_foliar_ = (δ^15^N-NH_4_^+^ × *f*_NH4_^+^ + δ^15^N-NO_3_^−^ × *f*_NO3_^−^ + δ^15^N-DON^−^ × *f*_DON_)(4)
1 = *f*_NH4_^+^ + *f*_NO3_^−^ + *f*_DON_(5)
where *f*_NO3_^−^ and *f*_DON_ respectively represented the proportions of NH_4_^+^ and NO_3_^−^ and DON to plant N sources, which were obtained by the ‘iso-source’ model calculation method [66].

#### 4.4.2. Estimation of Plants’ Preference for N Sources

If the contribution of one N form to the absorption of all N sources by plants is greater than the contribution of the specific N source to the total availability of all forms combined, it can be assimilated by plants first, which is reported as the N preference of plants [11,22]. The preference of dominant plants for NH_4_^+^ can be expressed as the difference between the contribution degree of NH_4_^+^ absorption to DIN absorption and the proportional contribution of NO_3_^−^ absorption to DIN, which was calculated as the following:*β*_NH4+_ = *f*_NH4+_/*f*_DIN_ − ([NH_4_^+^]/[DIN])(6)
where *f*_DIN =_ *f*_NH4_^+^ + *f*_NO3_^−^. [NH_4_^+^] and [DIN] were the mean of NO_3_^−^ and DIN content in soil at different stages. Positive, 0 and negative values of *β*_NH4_^+^ indicate that plants prefer NO_3_^−^, no preference, NH_4_^+^ [65].

Plant N uptake preference for DON was estimated using the following equations:*β*_DON_ = *f*_DON_/*f*_TDN_ − ([DON]/[TDN])(7)
where *f*_TDN_ = *f*_DIN_ + *f*_DON._ Positive, 0, and negative values of *β*_DON_ indicate plant preference DON, no preference, and plant preference DIN [22].

### 4.5. Statistical Analysis

After sorting preliminary data, One-way ANOVA was applied to compare whether there were significant differences in soil basic properties and N contents of different forms at each chronosequence stage (*p* < 0.05). We adopted the same one-way ANOVA to evaluate foliar N and soil N pool isotope (δ^15^N) values and compared the relative contribution percentages of soil NH_4_^+^, NO_3_^−^ and DON to dominant plants N uptake at each specific stage. Duncan test was used for *post-hoc* multiple comparisons if the *F* value was significant (*p* < 0.05). Linear regression analysis was produced to test the correlation between the contribution of different forms of N to the dominant plant N source and soil N pool size during primary succession. All data were statistically analyzed in SPSS version 16.0 (SPSS Inc., Chicago, IL, USA).

## 5. Conclusions

The ongoing conditions of global warming accelerate glacier retreats, especially in the last few decades. Understanding the patterns and drivers of ecosystem succession is a prerequisite for sustainable management in these fragile environments. Collectively, this study showed that soil inorganic nitrogen had always been the most important N source, and indicated a transition from NO_3_^−^ for dominant plants in the early stage to NH_4_^+^ in the later stage during primary succession in the Hailuogou glacier retreat of the eastern Tibetan Plateau, China. Dominant plants constantly adjust their N use strategies to adapt to environmental changes, which are jointly affected by the characteristics of the plant and soil nutritional status. Further studies will be necessary in the future, and it is helpful to reasonably evaluate the preference of N acquisition of dominant plants and reveal the response of N uptake preference of dominant plants during primary succession. The extrapolation of relationships between N source contributions and stand age must be considered with caution, and future long-term study is necessary to better understand plant N acquisition strategies and their association with plantation restoration.

## Figures and Tables

**Figure 1 plants-12-00838-f001:**
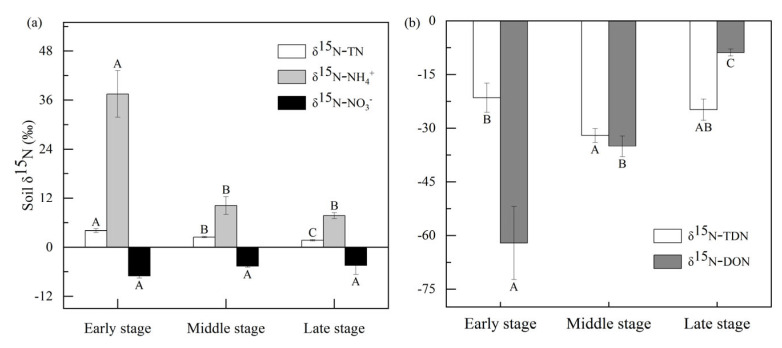
Soil δ^15^N values of DON, NH_4_^+^, NO_3_*^−^* (**a**) and TN, TDN (**b**) in soil at different successional stages. White, gray and black bars represent δ^15^N−TN, δ^15^N−NH_4_^+^, δ^15^N−NO_3_*^−^* δ^15^N−TDN and δ^15^N−DON values of soil, respectively. Each value is the ±1 SE of the mean and those followed by different letters in the same column are significantly different (*p* < 0.05) for dominant species at different successional stages (early (~12 year), middle (~40 year) and late (~120 year) stage). Different capital letters denote significant differences among successional stages, and different species (*p* < 0.05), according to Tukey’s test of one-way ANOVA.

**Figure 2 plants-12-00838-f002:**
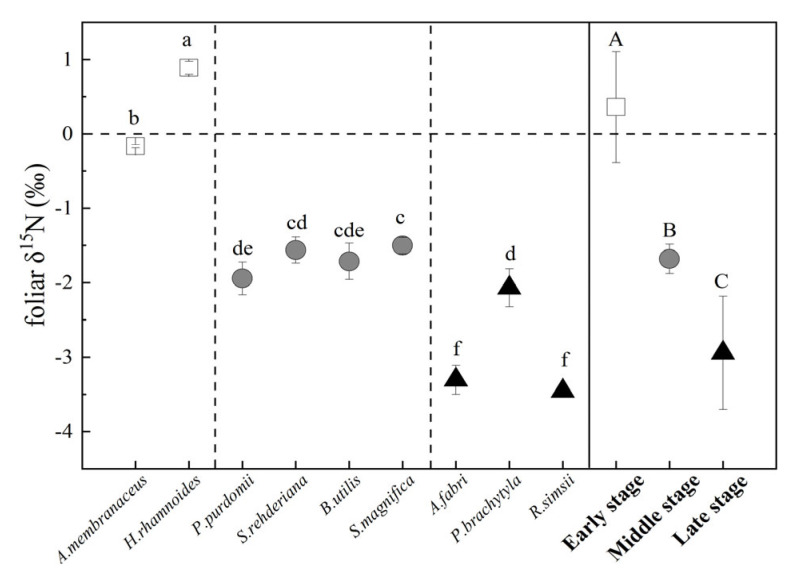
Foliar δ^15^N values of dominant species at different successional stages. White, gray, and black circles represent species from the early- (~12 year), middle- (~40 year), and late-(~120 year) succession stages, respectively. Different capital and lowercase letters denote significant differences among successional stages, and different species, respectively (*p* < 0.05), according to Tukey’s test of one-way ANOVA.

**Figure 3 plants-12-00838-f003:**
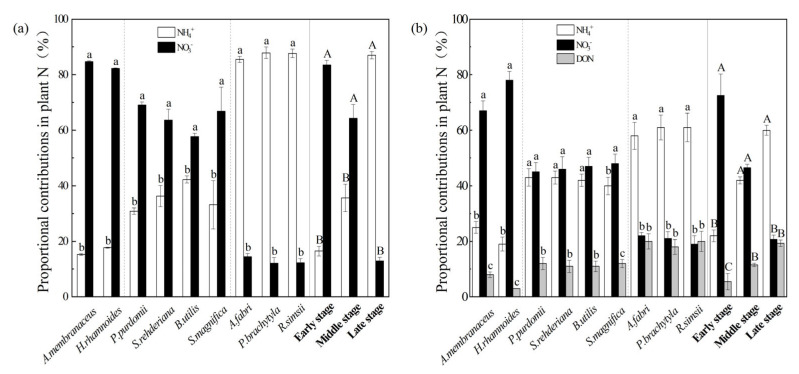
The proportional contributions (%) of soil NH_4_^+^ and NO_3_*^−^* (**a**) in dominant plants N, and the proportional contributions (%) of soil NH_4_^+^ and NO_3_*^−^* and DON (**b**) in dominant plants N at different successional stages. White, gray, and black bars represent the proportional contributions (%) of soil NH_4_^+^ and NO_3_*^−^* and DON in the plant nutrition of dominant species, respectively. Different capital and lowercase letters denote significant differences among successional stages, and different species, respectively (*p* < 0.05), according to Tukey’s test of one-way ANOVA.

**Figure 4 plants-12-00838-f004:**
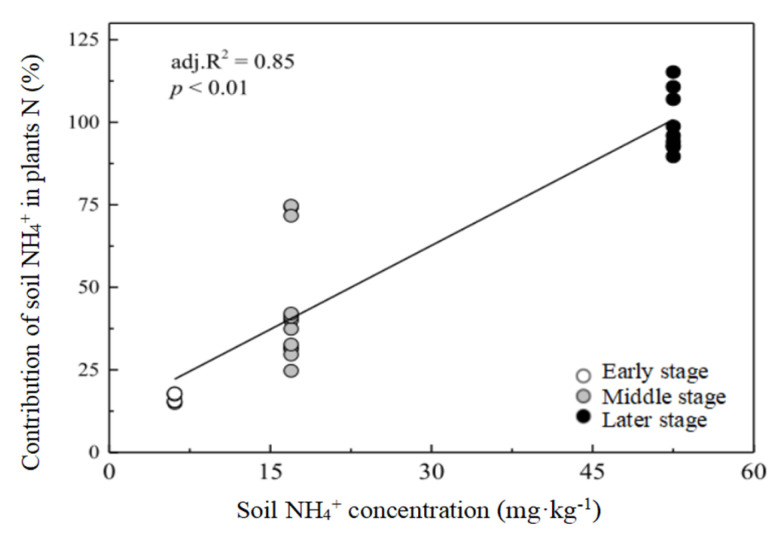
The relationships between soil NH_4_^+^ concentration (mg·kg^−1^) and the proportional contribution (%) of soil NH_4_^+^ in dominant plants N during primary succession. R^2^ denotes the proportion of variance explained.

**Figure 5 plants-12-00838-f005:**
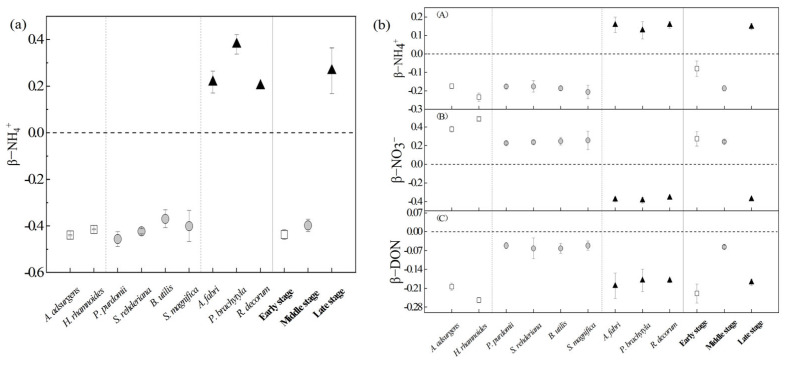
Preference (*β*) for one soil N source of dominant plants based on the two-source mixing model (**a**) and the three-source mixing model (**b**) at different successional stages. Positive *β* values denote a preference. Negative *β* values or *β* = 0 indicate no preference. White, gray, and black circles represent species from the early− (~12 year), middle− (~40 year), and late− (~120 year) succession stages, respectively. A, B and C mean preference for NH_4_^+^, NO_3_^−^ and DON, respectively.

**Table 1 plants-12-00838-t001:** Values of soil pH, contents of SOM (g·kg^−^^1^), NH_4_^+^, NO_3_*^−^*, DON, TDN (mg·kg^−^^1^) of at different successional stages.

Different Stages	Early Stage	Middle Stage	Late Stage
pH	6.50 ± 0.20 ^a^	5.97 ± 0.45 ^a^	4.73 ± 0.40 ^b^
SOM	1.50 ± 0.03 ^c^	2.30 ± 0.47 ^b^	3.50 ± 0.41 ^a^
NH_4_^+^	6.10 ± 1.31 ^c^	17.08 ± 1.04 ^b^	51.46 ± 3.45 ^a^
NO_3_^−^	4.20 ± 0.29 ^c^	11.93 ± 1.40 ^b^	19.41 ± 1.62 ^a^
DON	4.38 ± 0.56 ^b^	6.99 ± 1.02 ^b^	44.21 ± 5.23 ^a^
TDN	14.36 ± 0.15 ^b^	27.95 ± 0.89 ^b^	117.23 ± 19.69 ^a^

Each value is the ±1 SE of the mean and followed by different lowercase letters in the same line are significantly different (*p* < 0.05) for dominant species at different successional stages (early (~12 year), middle (~40 year) and late (~120 year) stage). Differences within or between successional stages were detected by One-Way ANOVA and followed by Duncan test if significances existed (*p* < 0.05). Abbreviations: SOM, soil organic matter; NH_4_^+^, ammonium; NO_3_^−^, nitrate; DON, dissolved organic nitrogen; TDN, total dissolved N.

**Table 2 plants-12-00838-t002:** Identities, types of mycorrhizal fungi and importance values of dominant species at different successional stages.

Different Stages	Dominant Species	Types of Mycorrhizal Fungi	Importance Values
Early stage	*Astragalus membranaceus*	N-fixing diazotroph	0.52
(~12 year)	*Hippophae rhamnoides*	N-fixing diazotroph	0.35
Middle stage	*Populus purdomii*	Arbuscular mycorrhiza	0.34
(~40 year)	*Salix rehderiana*	Arbuscular mycorrhiza	0.20
	*Betula utilis*	Ectomycorrhiza	0.13
	*Salix magnifica*	Arbuscular mycorrhiza	0.07
Late stage	*Abies fabri*	Ectomycorrhiza	0.40
(~120 year)	*Picea brachytyla*	Ectomycorrhiza	0.31
	*Rhododendron simsii*	Ericoid mycorrhiza	0.13

## Data Availability

The data presented in this study are available on reasonable request from the corresponding author.

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
