# Peer review of "Nitrogen Preference of Dominant Species during Hailuogou Glacier Retreat Succession on the Eastern Tibetan Plateau"

_plants, 2023, doi:10.3390/plants12040838_

Round 1

Reviewer 1 Report

The global warming accelerates the glacier melting. Therefore, it is an interesting topic to study nitrogen preference of dominant species during the glacier retreat succession. However, there are some issues that need revision or clarification.

(1) It seems that the conclusion of this study is different with common knowledge of N preference at the different ecosystem, i.e., dominant plants preferred to absorb NO3 - in the early (73.5%) and middle (46.5%) stages; at the late stage, soil NH4 + was preferred.  It is usually considered that amino acid and NH4 are the main form of available N, due to the poor mineralization. Authors also mentioned, "In N-restricted ecosystems such as Arctic tundra, boreal forest, alpine meadow, temperate forest, and temperate grassland, plants have a preference for soil amino acids even higher than inorganic N". I'm not expert on δ15N method. Please double check the method used.

(2) The NH4 contents in soil at the early and middle stages were higher than NO3 in soil; and at the late stage, NO3 was significantly higher than NH4. Why plants preferred NO3 at at the early and middle stages, and NH4 at the late stage?

(3) Please clarify whether the M&M section: Whether there are some differences in elevations of the different successional stages. If the pictures of the different successional stages could be provided, it would be helpful for understanding the study sites.

(4) N deposition rate at the study sites? How to measure the soil bulk density with disturbed soil sample (line 355)?

Author Response

Review 1

Comments and Suggestions for Authors:

  • It seems that the conclusion of this study is different with common knowledge of N preference at the different ecosystem, i.e., dominant plants preferred to absorb NO3- in the early (73.5%) and middle (46.5%) stages; at the late stage, soil NH4+ was preferred.  It is usually considered that amino acid and NH4+ are the main form of available N, due to the poor mineralization. Authors also mentioned, "In N-restricted ecosystems such as Arctic tundra, boreal forest, alpine meadow, temperate forest, and temperate grassland, plants have a preference for soil amino acids even higher than inorganic N". I'm not expert on δ15N method. Please double check the method used.

Thank you for your comments. Firstly, in the Hailuogou Glacier Chronosequence, the relatively mild and humid conditions promoted the rapid colonization of vegetation and soil development, also led to greater microbial turnover and exoenzyme activities, thus, higher mineralization rate and accumulation of various N components, according to our previous studies (Jiang et al. 2018; 2019; Lei et al. 2015; 2021). Secondly, Chapin et al. (1993) found that, in N-restricted ecosystems, especially Arctic tundra, plants prefer organic N forms including amino acids even though higher soil inorganic N concentrations, also proved by many later studies. Furthermore, we have double checked all the methods used in the manuscript, which were correct and commonly adopted.

Chapin, F.S.; Moilanen, L.; Kielland, K. Preferential use of organic nitrogen for growth by a non-mycorrhizal arctic sedge. Nature 1993, 361, 150-153. doi: 10.1038/361150a0.

Jiang, Y.L.; Lei, Y.B.; Wei, Q.; Korpelainen, H.; Li, C.Y. Revealing microbial processes and nutrient limitation in soil through ecoenzymatic stoichiometry and glomalin-related soil proteins in a retreating glacier forefield. Geoderma 2019, 338, 313-324. doi:10.1016/j.geoderma.2018.12.023

Jiang, Y.L.; Lei, Y.B.; Yang, Y.; Korpelainen, H.; Niinemets, U.; Li, C.Y. Divergent assemblage patterns and driving forces for bacterial and fungal communities along a glacier forefield chronosequence. Soil Biology & Biochemistry, 2018, 118, 207-216. doi:10.1016/j.soilbio.2017.12.019

Lei, Y.B.; Zhou, J.; Xiao, H.F.; Duan, B.L.; Wu, Y.H.; Korpelainen, H.; Li, C.Y. Soil nematode assemblages as bioindicators of primary succession along a 120-year-old chronosequence on the Hailuogou Glacier forefield, SW China. Soil Biology & Biochemistry, 2015, 88, 362-371. doi:10.1016/j.soilbio.2015.06.013

Lei, Y.B.; Du, L.S.; Chen, K.; Plenković-Moraj, A.; Sun, G. Optimizing foliar allocation of limiting nutrients and fast‐slow economic strategies drive forest succession along a glacier retreating chronosequence in the eastern Tibetan Plateau. Plant Soil, 2021, 462, 159-174. doi:10.1007/s11104-020-04827-3

  • The NH4+ contents in soil at the early and middle stages were higher than NO3- in soil; and at the late stage, NO3- was significantly higher than NH4+. Why plants preferred NO3- at the early and middle stages, and NH4+ at the late stage?

Thanks for your insightful comments. Plant N preferences depend on a wide and dynamic range of biotic and abiotic factors that overlap simultaneously, including abiotic environmental conditions, and soil nutrient status, as well as biotic physiological changes. Although the reasons for apparent N preference switches are currently poorly understood, one possible underlying mechanism may be the vegetation ontogeny and developmental stages. Because the toxicity caused by NH4+ accumulation in plant tissues will be harmful to tree growth, especially in young trees (Boudsocq et al. 2012), thus, plants at the early and middle stages which are usually young seedlings preferred NO3- to avoid possible NH4+ toxicity. Nevertheless, many other reasons may warrant further investigations.

Boudsocq, S.; Niboyet, A.; Lata, J.C.; Raynaud, X,; Loeuille, N.; Mathieu, J.; Blouin, M.; Abbadie, L.; Barot, S. Plant preference for ammonium versus nitrate: a neglected determinant of ecosystem functioning? The American Naturalist 2012, 180, 60-69. doi: 10.1086/665997

  • Please clarify whether the M&M section: Whether there are some differences in elevations of the different successional stages. If the pictures of the different successional stages could be provided, it would be helpful for understanding the study sites.

From the first (2982 m) to the last (2855 m) stage, there is only a 127 m elevational difference. Therefore, the spatial differences in the lithology, topography and climate are negligible in such small gradient in length (2 km), width (50-200 m) and the 127 m elevation. For the sampling locations picture, please refer to our previous paper (doi:10.1016/j.soilbio.2015.06.013).

  • N deposition rate at the study sites? How to measure the soil bulk density with disturbed soil sample (line 355)?

Song et al. (2017) found that the annual average bulk deposition fluxes of total dissolved nitrogen were 7.4 kg N ha−1 yr−1 at Gongga Mountain area during the period 2008 to 2013. The contributions of NH4+-N, NO3--N and DON to total N deposition were 57.8%, 28.8%, and 13.4%, respectively. We also added the detailed information for soil bulk density measurement in line 371 as “Soil bulk density (BD) was determined by the weight of the air-dried soil samples and the volume of the ring cutter (200 cm3)”.

Song, L.; Kuang, F.H.; Skiba, U.; Zhu, B.; Liu, X.J.; Levy, P.; Dore, A.; Fowler, D. Bulk deposition of organic and inorganic nitrogen in southwest China from 2008 to 2013. Environmental Pollution, 2017, 227, 157-166. doi: 10.1016/j.envpol.2017.04.031

Reviewer 2 Report

I found this study to be well designed, with clear results, interesting to read.

I do not have any specific comments to the experiment, methods and results.

Minor comments:

I miss source of data presented in tab. 2.

In Introduction, all the abbreviations are explained when they appear in the text for the first time, but not in Results (incl tables and figures).

Author Response

Review 2

I found this study to be well designed, with clear results, interesting to read.

I do not have any specific comments to the experiment, methods and results.

Response: Thank you for your positive comments.

Minor comments:

  • I miss source of data presented in tab. 2.

Response: Thank you for pointing out our typos. We have added source of data presented in tab. 2.

  • In Introduction, all the abbreviations are explained when they appear in the text for the first time, but not in Results (incl tables and figures).

Response: We have revised and explained all the abbreviations when they appear in the text for the first time.

Round 2

Reviewer 1 Report

Authors have responded my comments and suggestions accordingly.